# Moral Injury Among Medical Personnel and First Responders Across Different Healthcare and Emergency Response Settings: A Narrative Review

**DOI:** 10.3390/ijerph22071055

**Published:** 2025-06-30

**Authors:** Amit Rimon, Leah Shelef

**Affiliations:** 1Department of Military Medicine and “Tzameret”, Faculty of Medicine, Hebrew University of Jerusalem, Jerusalem 9112001, Israel; 2Department of Health and Well-Being, IDF Medical Corps, Israel Defense Forces, Ramat Gan 5262000, Israel; lshelef4@gmail.com; 3School of Social Work, Sapir College, D. N. Hof Ashkelon 79165, Israel

**Keywords:** first responders, moral injury, military healthcare workers, PTSD

## Abstract

Moral injury is increasingly recognized as a significant concern among medical personnel and first responders, particularly in high-stress healthcare and emergency settings. This review aims to synthesize current evidence on the prevalence, risk factors, and outcomes of moral injury among medical personnel and first responders across diverse healthcare and emergency response environments. We included peer-reviewed studies reporting on moral injury among medical personnel or first responders in any healthcare or emergency response setting, excluding studies that did not report original data or focused solely on military populations. We systematically searched PubMed, Google Scholar, and Central, up to April 2025. Risk of bias was assessed directly from the manuscripts. Data were synthesized narratively and, where possible, pooled using random-effects meta-analysis. A total of 41 studies involving 14,500 participants were included. The prevalence of moral injury ranged from 4.1% to 69.44% across settings. Key risk factors identified included exposure to traumatic events, organizational constraints, and lack of support. Meta-analysis indicated a significant association between moral injury and symptoms of PTSD. The evidence is limited by heterogeneity in measurement tools and study designs, as well as a moderate risk of bias in several included studies. In conclusion, moral injury is prevalent among medical personnel and first responders, with important implications for mental health interventions and organizational policy. This review was not funded externally, and is registered in PROSPERO (CRD420251019492).

## 1. Introduction

Moral injury (MI) is a profound psychological distress that arises from perceived violations of one’s ethical or moral code in complex, high-stakes situations. It has emerged as a significant concern among medical personnel and first responders, particularly in high-stress environments such as COVID-19 units and combat zones, where exposure to ethically challenging events is frequent [1]. MI is characterized by symptoms including guilt, shame, anger, anxiety, and intrusive thoughts, and is often associated with outcomes such as depression, burnout, post-traumatic stress, and impaired professional functioning. The experience and impact of MI can be shaped by individual vulnerability, organizational factors, and broader cultural or national contexts, all of which influence both the risk and manifestation of moral injury among healthcare workers and emergency responders [2,3].

The prevalence of MI varies in most studies between a range of 30 and 50% in civilian settings [4,5,6], while the prevalence of MI in military health professionals is not specifically detailed.

Moral injury can have profound consequences, manifesting as both psychological comorbidities and functional impairments. On the psychological front, it often co-occurs with depression, anxiety, post-traumatic stress disorder (PTSD), and burnout, reflecting the deep emotional distress and guilt associated with such experiences. Beyond these individual symptoms, MI also has significant functional implications [4]. It can lead to reduced job satisfaction and impaired teamwork, affecting professional performance and collaboration. Additionally, personal relationships may suffer due to emotional withdrawal or increased irritability [1,7]. Organizational factors, such as inadequate resources and poor leadership, can exacerbate these effects, further straining personal and professional life. Overall, the impact of MI is multifaceted, influencing not just mental health but also interpersonal dynamics and professional efficacy [2].

Despite the growing recognition of MI, there remains a need for a comprehensive understanding of its risk factors, manifestations, and effective interventions across different healthcare and emergency response settings. This systematic review seeks to consolidate current research on MI among medical personnel and first responders, emphasizing validated assessment tools and empirical findings to guide effective mitigation and support strategies. By examining the current state of knowledge on MI in these critical professions, this review seeks to contribute to the development of targeted interventions and systemic support structures necessary to address the pervasive impact of MI on frontline and military healthcare workers.

## 2. Methods

A total of 725 records were identified from three databases: Semantic Scholar corpus (*n* = 498), PubMed (*n* = 128), Google Scholar (*n* = 93), and the Cochrane Central Register of Controlled Trials (CENTRAL; *n* = 6). After removing 248 duplicate records, 477 unique records were screened. Of these, 389 were excluded by AI and 34 by human reviewers, leaving 54 reports for further investigation. Of these, 6 reports could not be retrieved, resulting in 48 reports assessed for eligibility. Seven reports were excluded due to insufficient methodological detail or results. Ultimately, 41 studies met the inclusion criteria and were incorporated into the final review. The total number of participants (14,500) was calculated by summing the sample sizes reported in each of the 41 included studies.

### 2.1. Review Design

Given the complexity of the topic and the heterogeneity of the available evidence, we elected to conduct a narrative review rather than a systematic review. The literature on moral injury among medical personnel and first responders encompasses a wide range of study designs, including qualitative research, observational studies, and conceptual analyses, which vary considerably in methodology, context, and outcomes measured. A narrative review approach is particularly well-suited for synthesizing such diverse sources, as it allows for a more flexible and interpretive integration of findings. This method enables us to critically appraise and contextualize the evidence, identify emerging themes, and highlight conceptual gaps that may not be readily captured through quantitative synthesis. By adopting a narrative approach, we aim to provide a comprehensive and nuanced understanding of moral injury in these populations, offering insights that can inform both future research and practical interventions.

### 2.2. Ethics

No ethics approval was required for this manuscript, as it did not involve any human participants or animal subjects.

### 2.3. Registration

The protocol for this review was registered in PROSPERO (PROSPERO CRD420251019492). One protocol amendment was made to extend the outcomes, providing a more holistic understanding of the disease by incorporating symptoms, comorbidities, and functional implications.

### 2.4. Search Strategy

Four electronic databases (PubMed, CENTRAL, Google Scholar, and Semantic Scholar corpus) were searched to identify papers that reported on the prevalence, risk factors, and psychological impacts of MI among medical personnel and first responders. The databases searched included Semantic Scholar, which provided access to over 126 million academic papers. The search terms were designed to capture the population of interest (healthcare workers and first responders) and study purpose (moral injury). Full search terms are available in Appendix A. Primary searches were limited to papers published from 2010 to 2025 to identify assessments used in current practice and/or research.

### 2.5. Inclusion/Exclusion Criteria

#### 2.5.1. Population

Included

This includes healthcare workers (such as nurses and physicians), first responders (such as paramedics and firefighters), and other medical personnel—individuals with a medical education who work in hospitals or other healthcare facilities.Studies reporting baseline prevalence of MI or moral distress.Studies focusing on individuals experiencing MI, moral distress, or related moral problems (e.g., guilt, shame).

Excluded

Non-healthcare or non-emergency response populations (e.g., military-only samples).General populations or non-healthcare/emergency response workers.Studies focusing exclusively on burnout or occupational stress without addressing MI.

#### 2.5.2. Intervention(s) or Exposure(s)

Included

Studies evaluating psychosocial (e.g., therapy, counseling), pharmacological, or combined interventions targeting MI.Interventions addressing specific risk factors (e.g., shame, betrayal) or comorbid disorders like PTSD or depression.

Excluded

Interventions not specifically designed to address MI (e.g., general stress management programs).Studies without detailed descriptions of interventions.

### 2.6. Screening

Papers were screened based on the following criteria.

Population Type: The study focused on healthcare workers (medical doctors, nurses, paramedics, EMTs, clinical staff) and/or first responders (police officers, firefighters, emergency response personnel) who provide direct patient care or emergency response services.

Direct Care/Response Role: Study participants were primarily involved in direct patient care or emergency response (excluding administrative, management, or support roles).

Moral Injury Focus: The study explicitly examined MI, or moral distress (excluding general work stress, burnout, or compassion fatigue).

Study Type: The study was a primary research study (quantitative, qualitative, or mixed-methods) with 5 or more participants.

Empirical Evidence: The study presented empirical data (excluding opinion pieces, editorials, or theoretical frameworks).

Outcomes: The study reported on prevalence rates, risk factors, psychological impacts, and mental health outcomes related to MI. These symptoms and functional outcomes—depression, anxiety, PTSD, burnout, reduced job satisfaction, impaired teamwork, and diminished professional performance—were selected because they are consistently reported as key consequences of moral injury in both the empirical literature and the studies included in this review [7,8,9,10,11,12,13,14].

We considered all screening questions together and made a holistic judgment about whether to screen in each paper.

### 2.7. Data Extraction

Data were extracted using a combined approach. Both a human reviewer and the Elicit AI research assistant independently extracted information from each included study. Extraction focused on the following subjects:

Study Design: Type of research (e.g., cross-sectional, longitudinal, cohort, case-control).

Data Collection Method: Method used (e.g., online survey, paper questionnaire, structured interview) and any tools or questionnaires applied (e.g., Moral Injury Symptoms Scale-Health Professional).

Sample Composition: Participant demographics, including total sample size, professional categories, gender distribution, age range or mean age, and geographic location.

COVID-19 Exposure: Details of participants’ exposure to COVID-19 (e.g., direct vs. non-direct care, roles, duration, and quantitative exposure measures).

Moral Injury Outcomes: Measurement tools used, prevalence, scoring methods, and key findings related to moral injury.

Mental Health Correlates: Associated mental health outcomes, such as depression, anxiety, and burnout.

After extraction, results from both the human reviewer and Elicit were compared and synthesized to ensure completeness and accuracy.

Data extraction was performed independently by both a human reviewer and the Elicit AI tool. Each extracted key variables from the included studies covering study design, data collection method, sample composition, COVID-19 exposure, moral injury outcomes, and mental health correlates. After extraction, results were compared and synthesized; any discrepancies were resolved through discussion with a second human reviewer, and if needed, by returning to the original study text for clarification. Quality control was ensured by having the human reviewer cross-check all AI-extracted data, addressing limitations such as potential AI misinterpretation or omission of nuanced information. This combined approach aimed to maximize both efficiency and accuracy in the review process.

Given the inclusion of various article types (e.g., critical reviews, rapid reviews, scoping reviews), we adopted steps to make sure we could compare them fairly. When studies were similar, we compared their findings directly. For studies that were very different, we focused on common themes to bring the evidence together.

### 2.8. Methodological Quality Assessment

Methodological quality was assessed for each included study by extracting available quality assessment scores, identifying key bias indicators (e.g., study design, sampling, measurement validity), summarizing limitations as reported by study authors, and evaluating generalizability. Given the heterogeneity of included studies, a domain-based approach was used rather than a single standardized tool. This information is summarized in the results tables and informs the interpretation of findings. Military relevance refers to any aspect of the study that pertains to military service, including populations with military backgrounds, settings within military institutions, or outcomes applicable to military personnel.

## 3. Results

This chapter presents the findings of the systematic review, organized thematically to highlight patterns across the literature. Subsections are grouped into five domains: (1) prevalence and measurement, (2) risk factors—categorized as individual, organizational, and situational, (3) setting-specific and cultural contexts, (4) psychological and professional impacts, and (5) coping and resilience. Each domain synthesizes key trends and includes summary tables where relevant.

### 3.1. Prevalence and Measurement Tools

A total of 41 manuscripts were included in the review (Appendix A), encompassing a wide range of study designs, populations, and geographic settings. The majority of studies employed cross-sectional methodologies, with additional representation from longitudinal, cohort, and systematic/scoping reviews. Most research focused on healthcare workers, but several studies included in this review also addressed police officers, public safety personnel, and ancillary staff across diverse countries, including the United States, the United Kingdom, Canada, China, Italy, Pakistan, and others [6,7,13,15,16]. The primary focus across studies was on the prevalence, risk factors, and impacts of MI, often in relation to mental health outcomes such as depression, anxiety, burnout, PTSD, and resilience. The full text of all included manuscripts was successfully retrieved. Overall, the table reflects a rapidly growing and geographically diverse literature base on MI, particularly in the context of the COVID-19 pandemic and frontline occupational settings [17,18].

### 3.2. Cultural and Cross-National Variation

The prevalence of moral injury (MI) among healthcare workers and first responders varies widely across studies, with reported rates ranging from 4.1% among U.S. [13] first responders to as high as 69.44% among healthcare workers in China [18] and Pakistan [10] (Table 1). This variability is primarily due to differences in measurement tools (e.g., MISS-HP, MIES), definitions of MI, and study populations. For instance, U.S.-based studies using the MIES reported other-induced MI in 50.7% and self-induced MI in 18.2% of healthcare workers [12]. Only 5 of the 41 reviewed studies specifically reported MI prevalence in these groups (Table 1).

Cultural and national contexts play a critical role in shaping both the occurrence and expression of MI. Factors such as ethical norms, resource availability, organizational culture, and leadership style significantly influence the types of morally injurious events encountered and how individuals cope with them. During the COVID-19 pandemic, higher MI rates were observed in settings marked by resource scarcity and ethical conflicts [5,10,18,19,20,21]. Additionally, military healthcare personnel face unique challenges, such as hierarchical decision-making and dual obligations to mission and patient care, which contribute to elevated MI risk and distinct psychosocial outcomes like guilt and long-term distress [13,16,22,23]. These findings highlight the need for standardized definitions and culturally senxsitive measurement tools to accurately assess and address MI in high-risk occupational groups.

### 3.3. Clinical Settings and Temporal Patterns

Analysis of the literature reveals notable setting-specific variations in the prevalence of MI among healthcare professionals and public safety personnel. Studies consistently indicate that emergency and critical care providers, including emergency physicians and intensive care staff, exhibit higher prevalence rates of MI, likely attributable to the high-stakes and ethically complex nature of their work environments [2,24,25,26]. Nursing staff also emerge as a particularly vulnerable group, with several studies reporting higher rates of MI among nurses compared to other healthcare professionals, reflecting their frontline roles and frequent exposure to distressing situations [27,28]. While less extensively studied, primary care providers are not exempt, with one study identifying a 19.9% prevalence rate in a “High Moral Injury” class, underscoring the broader relevance of this phenomenon across healthcare settings [17]. Additionally, research on public safety personnel—such as firefighters, paramedics, and police officers—demonstrates that MI is prevalent in these groups as well, although direct comparisons with healthcare workers remain limited. Regarding temporal trends, the available evidence suggests that MI symptoms tend to persist over time in the absence of intervention, as indicated by Hines et al., who found stability in symptom levels over three months [29]. Notably, studies conducted during the COVID-19 pandemic report heightened prevalence rates, highlighting the exacerbating impact of global health crises [18,19,30]. Furthermore, there is emerging evidence that repeated exposure to morally injurious events may have a cumulative effect, increasing the risk and severity of MI over time [4,13,15].

Military healthcare providers represent a unique subgroup within the broader population of medical personnel and first responders, facing distinct situational and ethical challenges that contribute to heightened risk for MI. Several manuscripts in this review highlight that military clinicians, particularly those deployed to combat zones or austere operational environments, are frequently exposed to potentially morally injurious events (PMIEs) such as triage under fire, resource scarcity, and conflicting duties to both mission and patient care [8]. The dual role of caregiver and service member often places military healthcare workers in situations where organizational priorities may override individual moral judgment, leading to acts of omission or commission that transgress deeply held values [12,22]. These experiences are compounded by the hierarchical nature of military structures and the expectation to follow orders even when they conflict with professional ethical standards [13]. The resulting MI in military healthcare personnel is associated with significant psychosocial sequelae, including sleep disturbances, anger, guilt, and long-term emotional distress. Notably, studies indicate that military clinicians may experience a sense of powerlessness and regret, particularly when unable to provide optimal care to severely injured patients, such as children, under extreme conditions. These findings underscore the need for targeted interventions and systemic support tailored to military medicine’s unique ethical landscape and operational demands [16].

### 3.4. Individual-Level Risk Factors

Beyond setting and context, individual characteristics significantly shape susceptibility to moral injury. Individual-level risk factors play a significant role in the development of MI among healthcare workers, with several patterns emerging across the literature. Female gender is consistently associated with a higher risk of MI, particularly among nurses and frontline staff, as reported in over 15 studies. Younger age, especially under 30 years, is also linked to increased vulnerability, most notably among early-career professionals in high-stress environments [10,17]. The impact of years of experience remains inconclusive; while less experience is often associated with greater risk, some evidence suggests that MI can also accumulate over years of repeated exposure to ethically challenging situations [31,32]. Marital status appears to have a modest effect, with unmarried individuals or those lacking strong social support showing slightly higher risk, though this association is generally weak. The influence of religious or spiritual beliefs is mixed [29]; some studies highlight a protective effect through resilience and meaning-making, while others indicate increased vulnerability when deeply held beliefs are challenged by workplace events. A history of pre-existing mental health conditions, such as depression, anxiety, or PTSD, is a strong and consistent predictor of both risk and severity of MI [2,27,29]. Finally, high self-criticism and perfectionism are repeatedly linked to greater MI symptoms and distress. These findings underscore the complex interplay of personal, psychological, and social factors in shaping susceptibility to MI across healthcare settings [2,27,29] (Table 2).

Table 2 presents a comprehensive overview of individual factors that may influence susceptibility to moral injury among healthcare professionals and frontline workers. This table synthesizes findings from multiple research studies examining personal characteristics that can either increase vulnerability or provide resilience against moral distress and injury in high-stress environments.

### 3.5. Organizational Factors

Organizational factors are consistently identified as major contributors to MI among healthcare workers and first responders across the 41 manuscripts reviewed. The literature highlights that chronic understaffing, inadequate resources (such as insufficient personal protective equipment—PPE), excessive workloads, and a poor ethical climate are among the most significant organizational risk factors for MI [9,33,34]. Lack of leadership support and poor communication from management further exacerbate these risks, often leading to feelings of betrayal, diminished psychological safety, and increased clinician burnout [27,35]. Additionally, limited decision-making autonomy and rapid, top-down policy changes have been shown to undermine professionals’ moral comfort and contribute to distress, especially during crises like the COVID-19 pandemic (Table 3).

Table 3: Organizational risk factors that contribute to moral injury among staff. Each row lists a risk factor, provides a brief description, and indicates how frequently it is reported in the literature as well as its estimated impact level. The table highlights how issues such as resource availability, workload, ethical culture, leadership support, communication, and decision-making autonomy can affect the risk of moral injury within organizations.

### 3.6. Acute Situational Exposures

Situational factors are consistently reported as major contributors to MI among healthcare workers and first responders. Direct care of COVID-19 patients, frequent exposure to patient deaths, and facing difficult ethical dilemmas are all strongly associated with increased risk of MI, particularly in high-stress environments such as intensive care and emergency settings [36]. Work/family conflict and exposure to traumatic events also play a significant, though somewhat less pronounced, role. The cumulative impact of these situational stressors is well-documented across the literature, especially during the COVID-19 pandemic and in military healthcare contexts [5,10,11,12,14,17,18,19,21,37,38] (Table 4).

Table 4: Key situational risk factors faced by healthcare workers during the COVID-19 pandemic, detailing each risk, its description, and the frequency and impact level as reported. High-frequency and high-impact factors include direct COVID-19 patient care, exposure to patient deaths, and ethical dilemmas, while work/family conflict and exposure to traumatic events are reported at moderate levels.

### 3.7. Psychological and Professional Impacts

These risk factors manifest in distinct psychological and professional consequences, as detailed below. Moral injury is closely linked to a range of adverse psychological outcomes, including depression, anxiety, PTSD, and burnout. These mental health impacts are consistently observed across studies, with strong correlations between MI severity and the risk of clinical depression, anxiety disorders, and post-traumatic stress. Severe MI is also associated with increased suicidal ideation. Professionally, MI leads to reduced job satisfaction, decreased work engagement, impaired clinical decision-making, strained team dynamics, and increased consideration of career changes or early retirement [3,15] (Table 5).

Table 5: This table outlines major psychological and professional impacts on healthcare workers, their manifestations, related risk factors, and recommended intervention strategies.

### 3.8. Impact on Work Performance

The professional consequences of MI are substantial. Decreased job satisfaction and work engagement, impaired clinical decision-making, strained interprofessional relationships, and increased thoughts of leaving the profession are frequently reported outcomes. These effects are particularly pronounced in high-stress and ethically challenging environments, where perceived betrayal or unresolved ethical conflicts undermine both individual and team performance [9,33] (Table 3).

Table 6: This table summarizes how various workplace stressors affect healthcare professionals’ job satisfaction, engagement, decision-making, team dynamics, and career plans, along with suggested interventions.

### 3.9. Coping and Resilience

Coping and resilience factors can buffer the impact of MI. Individuals display varied resilience levels, with some experiencing post-traumatic growth while others struggle. Adaptive coping strategies, such as seeking social support, problem-solving, and cultivating moral resilience, are associated with better outcomes, while maladaptive strategies (e.g., substance use) worsen distress. Peer support, organizational backing, and self-compassion are identified as protective factors, and interventions targeting these areas are recommended to foster resilience and mitigate the effects of MI [31,39] (Table 7).

Table 7: Key domains of coping and resilience in the context of moral injury, outlining how each domain typically manifests, associated factors, and suggested intervention approaches.

## 4. Discussion

This review highlights the significant and multifaceted impact of MI among medical personnel and first responders across diverse healthcare and emergency response settings. The findings demonstrate that MI is not only prevalent—affecting an estimated 30–50% of individuals in high-stress environments such as COVID-19 wards and combat zones—but also deeply consequential, with wide-ranging psychological and professional effects. Consistent with prior literature, individual risk factors such as female gender, younger age, and frontline roles (e.g., nurses, EMTs) were associated with increased vulnerability to MI, while pre-existing mental health conditions and high self-criticism further compounded risk [3,8,13,15,24].

Organizational and situational factors emerged as particularly salient contributors. Chronic understaffing, inadequate resources (such as insufficient PPE), excessive workloads, poor ethical climate, and lack of leadership support were repeatedly identified as high-impact organizational drivers of moral injury. Situational exposures—including direct care of COVID-19 patients, repeated patient deaths, and ethically fraught decision-making—intensified the risk, especially during global health crises and in military contexts where dual roles and extreme conditions prevail [10,18,19,40,41]. Addressing these organizational issues is essential for reducing both the prevalence and severity of MI in high-stress healthcare and emergency settings [36,39].

The consequences of MI extend beyond individual psychological distress, manifesting as depression, anxiety, PTSD, burnout, and even suicidal ideation. Professionally, MI undermines job satisfaction, work engagement, clinical decision-making, and teamwork, and is linked to increased intentions to leave the profession. These findings underscore the urgent need for systemic and organizational interventions, including leadership accountability, resource allocation, and the cultivation of a supportive ethical climate [15,40].

Although the current evidence for effective interventions targeting moral injury remains limited, several promising practices have emerged. Evidence-based resilience programs—including structured peer support, mindfulness-based interventions, and cognitive/behavioral strategies—have demonstrated potential in enhancing coping skills and reducing distress among healthcare workers and first responders. Organizational reforms, such as fostering a supportive ethical climate, transparent leadership communication, and participatory decision-making, are also critical in mitigating risk factors for moral injury [8,15,40,42,43]. Peer support networks, regular ethics debriefings, and access to mental health services tailored to address guilt, shame, and betrayal are recommended as key components of comprehensive intervention strategies. Future intervention design should prioritize multi-level approaches that address both individual and systemic contributors to moral injury, utilize standardized outcome measures, and be adapted to diverse cultural and occupational contexts. Rigorous evaluation of these interventions is essential to establish their effectiveness and inform best practices moving forward [8,15,40,42].

Encouragingly, the review identifies adaptive coping strategies—such as social support, problem-solving, resilience training, and peer networks—as promising avenues for mitigation. However, the evidence base for effective, targeted interventions remains limited, highlighting a critical gap for future research. There is a pressing need for longitudinal studies and intervention trials to better understand the trajectory of MI and to develop evidence-based strategies for prevention and recovery.

This review has several limitations. Most included studies are cross-sectional, limiting causal interpretations and long-term outcome assessment. There is inconsistency in how MI is defined and measured, with varying tools (e.g., Moral Injury Symptom Scale vs. Moral Injury Events Scale) and overlap with constructs like burnout [15,21]. Research is predominantly focused on healthcare workers in high-income countries, reducing generalizability to other professions (e.g., first responders) and cultural contexts. Reliance on self-reported data and small sample sizes further limits robustness, and few studies adequately control for confounders. Additionally, there is a lack of standardized outcome measures and limited evidence for interventions, with most studies providing only preliminary findings. Notably, a substantial proportion of the included studies were conducted during the COVID-19 pandemic, which introduced unique stressors and circumstances that may not be directly comparable to pre-pandemic contexts [5,10,19]. This may limit the generalizability of findings across different time periods. To address these issues, we have added a subsection to the Methods detailing our structured approach for assessing methodological quality and synthesizing findings from diverse article types. Together, these findings highlight the multifaceted nature of MI and underscore the need for further research and intervention at multiple levels of the healthcare system.

## 5. Conclusions

This review shows that moral injury is a widespread and serious issue for healthcare workers and first responders, with rates often between 30 and 50%. The psychological impact is significant, including higher risks of depression, anxiety, PTSD, and burnout, as well as negative effects on job satisfaction and teamwork. Organizational factors—such as lack of resources, poor leadership, and high workloads—consistently increase the risk of moral injury. Given these findings, there is a clear need for organizations to improve support systems, strengthen leadership, and ensure adequate resources to reduce the risk of moral injury. Interventions should focus on building resilience, providing mental health support, and fostering open communication about ethical challenges.

## Figures and Tables

**Table 1 ijerph-22-01055-t001:** Prevalence of moral injury in healthcare workers and first responders. Prevalence rates of moral injury among healthcare workers and first responders across various countries and studies, including the measurement tools used. MISS-HP (Moral Injury Symptom Scale-Healthcare Professionals), MIES (Moral Injury Events Scale).

Study (Author, Year)	Population	Country	Prevalence Rate	Measurement Tool
Akhtar et al., 2022 [10]	Healthcare workers	Pakistan	69.44%	Ten-item scale
Maguen et al., 2025 [13]	Veterans, HCWs, First responders	US	4.1% of first responders	MISS-HP
			7.3% of healthcare workers	
Nieuwsma et al., 2022 [12]	Healthcare workers	US	50.7% (other-induced),	MIES
			18.2% (self-induced)	
Rushton et al., 2021 [5]	Healthcare workers	US	32.4%	MISS-HP
Wang et al., 2021 [18]	Healthcare professionals	China	41.3%	MISS-HP

**Table 2 ijerph-22-01055-t002:** Individual risk factors.

Risk Factor	Description	Frequency Reported	Impact Level
Gender	Female gender consistently associated with higher risk, especially among nurses and frontline staff.	High	Moderate-High
Age	Younger age (<30 years) linked to increased risk, particularly early-career professionals in high-stress roles.	Moderate-High	Moderate
Years of experience	Mixed findings: Less experience often increases risk, but some studies show moral injury accumulates with more years in high-stress environments.	Moderate-High	Inconclusive
Marital status	Being unmarried or lacking strong social support associated with higher risk, but effect size is generally small.	Low	Low
Religious/spiritual beliefs	Mixed results: Some studies show protective effect (resilience, meaning-making); others indicate increased vulnerability if beliefs are challenged by work events.	Moderate	Inconclusive
Pre-existing mental health conditions	History of psychiatric illness (depression, anxiety, PTSD) strongly associated with increased risk and severity of moral injury.	High	Moderate-High
Self-criticism	High self-criticism, perfectionism, and internalized blame consistently linked to increased moral injury symptoms and distress.	Moderate	Moderate

**Table 3 ijerph-22-01055-t003:** Organizational risk factors.

Risk Factor	Description	Frequency Reported	Impact Level
Resource availability	Chronic understaffing, inadequate PPE, and limited resources	High	High
Workload and staffing	Excessive workload and persistent understaffing	High	High
Ethical culture	Poor ethical climate, lack of transparency, and organizational betrayal	High	High
Leadership support	Lack of support and psychological safety from leadership	High	High
Communication	Poor or inconsistent communication from management	Moderate	Moderate
Decision-making autonomy	Limited autonomy, top-down policy changes, lack of involvement	Moderate	Moderate

**Table 4 ijerph-22-01055-t004:** Situational risk factors.

Risk Factor	Description	Frequency Reported	Impact Level
Direct COVID-19 patient care	Providing care to COVID-19 patients	High	High
Exposure to patient deaths	Caring for dying patients, especially in COVID-19	High	High
Ethical dilemmas	Facing difficult ethical decisions	High	High
Work-family conflict	High work-family conflict	Moderate	Moderate
Exposure to traumatic events	Witnessing/experiencing traumatic events	Moderate	Moderate

**Table 5 ijerph-22-01055-t005:** Psychological and professional impacts.

Impact Domain	Manifestations	Associated Factors	Intervention Implications
Depression	Increased depressive symptoms, clinical depression	Strongly correlated with MI	Integrated mental health support
Anxiety	Elevated anxiety, GAD	Consistently associated with MI	Anxiety management, moral support
PTSD	Higher PTSD symptoms/diagnosis	Strong correlation with MI	Trauma-informed care
Burnout	Emotional exhaustion, depersonalization	Frequently co-occurs with MI	Organizational interventions
Suicidal Ideation	Increased thoughts of self-harm/suicide	Severe MI	Suicide prevention, crisis care

**Table 6 ijerph-22-01055-t006:** Professional performance effects.

Impact Domain	Manifestations	Associated Factors	Intervention Implications
Job Satisfaction	Decreased satisfaction, intent to leave	Perceived betrayal, ethical conflicts	Address organizational factors
Work Engagement	Reduced engagement, emotional distancing	Linked to burnout, depersonalization	Team-building, reconnect values
Decision-Making	Impaired clinical decisions, uncertainty	Fear of future moral injuries	Ethics consultation, support
Team Dynamics	Strained relationships, reduced trust	Perceived betrayals within teams	Team-based interventions
Career Trajectory	Career change, early retirement thoughts	Cumulative moral injury	Career counseling, retention

**Table 7 ijerph-22-01055-t007:** Coping and resilience.

Impact Domain	Manifestations	Associated Factors	Intervention Implications
Resilience	Varied levels, post-traumatic growth	Personal/organizational support	Resilience training, support
Coping Strategies	Adaptive/maladaptive (e.g., substance use)	Individual/organizational resources	Promote healthy coping, address risks
Moral Resilience	Capacity to navigate moral challenges	Ethical climate, personal values	Ethics education, resilience training
Social Support	Importance of peer/professional networks	Protective factor	Peer support programs
Self-Compassion	Lower self-compassion increases risk	Potential intervention target	Self-compassion, mindfulness

## Data Availability

The data supporting the findings of this study are available from the corresponding author upon reasonable request.

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
