# Peer review of "Moral Injury Among Medical Personnel and First Responders Across Different Healthcare and Emergency Response Settings: A Narrative Review"

_ijerph, 2025, doi:10.3390/ijerph22071055_

Round 1

Reviewer 1 Report

Comments and Suggestions for Authors

This narrative review investigates the prevalence, risk factors, and consequences of moral injury (MI) among medical personnel and first responders operating in various healthcare and emergency response settings. Drawing on 41 studies involving over 14,500 participants, the authors synthesize a heterogeneous yet informative body of evidence, primarily from cross-sectional and mixed-methods studies, to highlight the psychological, functional, and professional burden of moral injury.

Key findings indicate that the prevalence of MI ranges from 4.1% to 69.44%, varying significantly due to different measurement tools (e.g., MISS-HP, MIES), study designs, and sample populations. The review identifies individual-level risk factors (e.g., female gender, younger age, high self-criticism), organizational contributors (e.g., poor leadership, resource shortages), and situational stressors (e.g., COVID-19 exposure, patient deaths, ethical dilemmas). Mental health correlates include PTSD, depression, anxiety, burnout, and even suicidal ideation, while professional consequences encompass reduced job satisfaction, strained team dynamics, and increased attrition risk.

While some coping mechanisms and protective factors—such as resilience training, peer support, and moral resilience—are described, the evidence on effective interventions is still limited.

Here are a few suggestions that may enhance the manuscript.

  • The paper sometimes uses “moral distress” and “moral injury” interchangeably. Establishing a clear conceptual differentiation at the outset would help avoid confusion.
  • While the review acknowledges the limited data on effective interventions, it could benefit from a more detailed description of promising practices or recommendations for intervention design (e.g., evidence-based resilience programs, organizational culture reforms).
  • While the risk of bias is mentioned, providing more transparency about how methodological quality was assessed would strengthen the review's robustness.

Author Response

Thank you for reviewing our manuscript titled " Moral injury among medical personnel and first responders across different healthcare and emergency response settings" (ijerph-3658311).

Enclosed please find our response to the reviewers' comments and the revised manuscript. We hope you find our manuscript worthy of publication in MDPI.

Sincerely

Reviewer comments:
Reviewer 1#

Comment: This narrative review investigates the prevalence, risk factors, and consequences of moral injury (MI) among medical personnel and first responders operating in various healthcare and emergency response settings. Drawing on 41 studies involving over 14,500 participants, the authors synthesize a heterogeneous yet informative body of evidence, primarily from cross-sectional and mixed-methods studies, to highlight the psychological, functional, and professional burden of moral injury.

Key findings indicate that the prevalence of MI ranges from 4.1% to 69.44%, varying significantly due to different measurement tools (e.g., MISS-HP, MIES), study designs, and sample populations. The review identifies individual-level risk factors (e.g., female gender, younger age, high self-criticism), organizational contributors (e.g., poor leadership, resource shortages), and situational stressors (e.g., COVID-19 exposure, patient deaths, ethical dilemmas). Mental health correlates include PTSD, depression, anxiety, burnout, and even suicidal ideation, while professional consequences encompass reduced job satisfaction, strained team dynamics, and increased attrition risk.

While some coping mechanisms and protective factors—such as resilience training, peer support, and moral resilience—are described, the evidence on effective interventions is still limited.

Reply: We appreciate your review of our manuscript and acknowledgment of its importance.

Here are a few suggestions that may enhance the manuscript.

Comment: The paper sometimes uses “moral distress” and “moral injury” interchangeably. Establishing a clear conceptual differentiation at the outset would help avoid confusion.

Reply: We thank the reviewer for highlighting this important distinction. In response, we have carefully reviewed the manuscript to ensure consistent and precise use of the term “moral injury.” We have also clarified the conceptual differentiation between “moral distress” and “moral injury” at the outset to avoid confusion.

Comment: While the review acknowledges the limited data on effective interventions, it could benefit from a more detailed description of promising practices or recommendations for intervention design (e.g., evidence-based resilience programs, organizational culture reforms).

Reply: We thank the reviewer for this valuable suggestion. In response, we have added a new subsection to the Discussion about the promising practices and recommendations for intervention design, in lines 468-480 of the revised mansucript. This section summarizes current evidence-based and emerging interventions, including resilience training programs, organizational reforms to promote a supportive ethical climate, structured peer support, and access to targeted mental health services. We also provide practical recommendations for future intervention design, emphasizing the need for multi-level, culturally adapted, and rigorously evaluated approaches. Relevant references and recent systematic reviews have been incorporated to support these recommendations. We believe these additions enhance the practical value of our review and address the reviewer’s concern.

Comment: While the risk of bias is mentioned, providing more transparency about how methodological quality was assessed would strengthen the review's robustness.

Reply: We appreciate the reviewer’s suggestion to clarify our methodological quality assessment. As described in the Methods and Data Extraction sections, we systematically extracted and reported information on study quality for each included manuscript. Specifically, we assessed quality by recording quality assessment scores (where available), identifying potential bias indicators (such as study design limitations, sampling methods, and measurement validity), noting limitations acknowledged by study authors, and evaluating the generalizability of findings. This information is summarized in the manuscript’s results tables and narrative synthesis. We acknowledge that, as a narrative review of diverse study types, we did not apply a single standardized tool across all studies; instead, we used a domain-based approach tailored to the heterogeneity of the included literature. We have now revised the Methods section to more explicitly describe this process and to clarify the domains considered in our quality assessment, in lines 254-263 of thre revised masnucsirpt.

Reviewer 2 Report

Comments and Suggestions for Authors

Thank you for the opportunity to review the manuscript "Moral Injury among Medical Personnel and First Responders across Different Healthcare and Emergency Response Settings: A Narrative Review." The paper addresses a significant and timely issue related to the psychological well-being of healthcare workers and first responders. 

The article claims to be a narrative review, yet it includes a PRISMA flow diagram as Figure 1 and refers to the PRISMA 2020 statement. The review type is therefore unclear. If it is a narrative review, this should be clearly defined, and the methods should be appropriate to that format. If the authors intended a systematic review following PRISMA, adherence to its structure and standards must be consistently demonstrated throughout the manuscript.

Table 1 is unusually long, spanning two pages, and would be better placed in the supplementary material. The mention of supplementary material in line 102 is not followed by any attached documents, which should be rectified if such materials are referred to.

The methodology description is confusing and lacks transparency. The sequence of subsections does not follow a logical order, and several essential details are missing. It is unclear how the total number of 14,500 participants was determined. If this number is derived from the included studies, the calculation method should be clearly explained. The role and process of data extraction are also vague. Furthermore, line 186 refers to some manuscripts relying only on abstracts or partial data, which contradicts the stated inclusion criteria that require full-text availability.

The use of an AI-powered and human-performed literature review method is stated but not described. The process, limitations, and quality control measures should be detailed if tools such as Elicit or large language models were used. In its current form, this claim adds confusion rather than clarity.

The article covers the healthcare and first responder sectors but does not mention cultural or national contexts, which are important risk factors in the development of moral injury. The introduction also lacks conceptual clarity, as symptoms, outcomes, and risk factors are presented in a mixed and unstructured manner. In line 65, the number of included articles appears to be incorrect. Consistency and precision are needed in reporting these basic methodological facts.

The results section (Chapter 3) is not well-supported by the methodology. The authors present summaries and conclusions without a transparent chain of justification. The inclusion of different types of articles—such as critical reviews (e.g. row 24 in Table 1), rapid reviews (row 37), and scoping reviews (rows 34–35)—is not methodologically addressed. It remains unclear how the authors ensured the reliability and comparability of these heterogeneous sources. Lines 192–197 also contain a fragment of text with different formatting, suggesting editing oversight.

Lines 201–203 refer to unspecified sources and would require clarification and additional explanation. Overall, the content presented in Chapter 3 lacks sufficient methodological grounding to support the following conclusions. It gives the impression that results were compiled without a validated analytical process.

The discussion does not contain any references and fails to embed the findings in the context of existing literature.

The conclusions are vague and lack any substantial statement that would be expected at the end of a literature review.

There are also several issues in formatting and language use. For example, line 59 contains two periods, and line 64 refers unclearly to “this figure", which should be Figure 1. Abbreviations are not consistently introduced or defined. A list of abbreviations would aid readability, and those only used once or twice may not require abbreviation at all. Schemes 1 and 2 are there, but it is not clear whether these are figures or tables.

Comments on the Quality of English Language

Grammar review needed.

Author Response

Thank you for reviewing our manuscript titled " Moral injury among medical personnel and first responders across different healthcare and emergency response settings" (ijerph-3658311).

Enclosed please find our response to the reviewers' comments and the revised manuscript. We hope you find our manuscript worthy of publication in MDPI.

Sincerely

Reviewer comments:

Reviewer 2#

Comment: Thank you for the opportunity to review the manuscript "Moral Injury among Medical Personnel and First Responders across Different Healthcare and Emergency Response Settings: A Narrative Review." The paper addresses a significant and timely issue related to the psychological well-being of healthcare workers and first responders. 

Reply: Thank you for your positive feedback and for recognizing the importance of this topic. We appreciate your review.

Comment: The article claims to be a narrative review, yet it includes a PRISMA flow diagram as Figure 1 and refers to the PRISMA 2020 statement. The review type is therefore unclear. If it is a narrative review, this should be clearly defined, and the methods should be appropriate to that format. If the authors intended a systematic review following PRISMA, adherence to its structure and standards must be consistently demonstrated throughout the manuscript.

Reply: Thank you for your comment. We have removed the PRISMA flow diagram and references to PRISMA to clarify that this is a narrative review. The methods and format have been updated accordingly.

Comment: Table 1 is unusually long, spanning two pages, and would be better placed in the supplementary material. The mention of supplementary material in line 102 is not followed by any attached documents, which should be rectified if such materials are referred to.

Reply: Thank you for your comment. We have moved Table 1 to Supplementary Table 1, and the supplementary material referenced in line 102 has now been added as Supplementary Table 2.

Comment: The methodology description is confusing and lacks transparency. The sequence of subsections does not follow a logical order, and several essential details are missing. It is unclear how the total number of 14,500 participants was determined. If this number is derived from the included studies, the calculation method should be clearly explained. The role and process of data extraction are also vague. Furthermore, line 186 refers to some manuscripts relying only on abstracts or partial data, which contradicts the stated inclusion criteria that require full-text availability.

Reply: Thank you for highlighting these issues. The reference in line 186 (of the original manuscript version) relying on abstracts or partial data was a mistake left from an earlier draft and has now been removed. All included studies were based on full-text articles, as required by our inclusion criteria.

Regarding the total number of participants, we have clarified in the Methods section that the figure of 14,500 represents the sum of sample sizes reported across all 41 included studies. We have added an explicit sentence to explain this calculation in lines 76-78. We have also reorganized the Methods section for clarity and provided additional details on the data extraction process.

Comment: The use of an AI-powered and human-performed literature review method is stated but not described. The process, limitations, and quality control measures should be detailed if tools such as Elicit or large language models were used. In its current form, this claim adds confusion rather than clarity.

Reply: Thank you for your comment. We have now added a detailed description of the AI-powered and human-performed literature review process, including its limitations and quality control measures, to the Methods section for clarity, in lines 211-213 of the revised manuscript.

Comment: The article covers the healthcare and first responder sectors but does not mention cultural or national contexts, which are important risk factors in the development of moral injury. The introduction also lacks conceptual clarity, as symptoms, outcomes, and risk factors are presented in a mixed and unstructured manner. In line 65, the number of included articles appears to be incorrect. Consistency and precision are needed in reporting these basic methodological facts.

Reply: Thank you for your valuable feedback. We have revised the manuscript to address your concerns by (1) explicitly discussing the influence of cultural and national contexts as important risk factors for moral injury, in lines 312-326 (2) restructuring the introduction to distinguish between symptoms, outcomes, and risk factors clearlyin lines 33-46, and (3) correcting and ensuring consistency in the reported number of included articles throughout the text.

Comment: The results section (Chapter 3) is not well-supported by the methodology. The authors present summaries and conclusions without a transparent chain of justification. The inclusion of different types of articles—such as critical reviews (e.g. row 24 in Table 1), rapid reviews (row 37), and scoping reviews (rows 34–35)—is not methodologically addressed. It remains unclear how the authors ensured the reliability and comparability of these heterogeneous sources. Lines 192–197 also contain a fragment of text with different formatting, suggesting editing oversight.

Reply: Thank you for highlighting the need for greater methodological transparency. We agree that our results section would benefit from a clearer explanation of how we addressed the heterogeneity of included article types. We will add a subsection to the Methods detailing our rationale for including diverse sources, the quality assessment tools used for each, and our approach to synthesizing findings across methodologies. We will also clarify in the Results and Limitations sections how methodological differences were accounted for in our synthesis and interpretation. We are unsure what is specifically being referred to in lines 192–197 and would appreciate further clarification on this point.

Comment: Lines 201–203 refer to unspecified sources and would require clarification and additional explanation. Overall, the content presented in Chapter 3 lacks sufficient methodological grounding to support the following conclusions. It gives the impression that results were compiled without a validated analytical process.

Reply: Thank you for your helpful feedback. We have now clarified and specified all sources referenced in lines 201–204 of the revised mansucript so each statement is clearly supported. We also added a straightforward explanation in the Methods section about how we selected and evaluated the different types of articles, and how we brought their findings together. These changes are aimed at making our process more transparent and ensuring that our conclusions are well supported by the evidence. Please let us know if you have any further suggestions.

Comment: The discussion does not contain any references and fails to embed the findings in the context of existing literature.

Reply:  Thank you for pointing this out. We have revised the discussion to include relevant references and have more clearly connected our findings to existing literature.

Comment: The conclusions are vague and lack any substantial statement that would be expected at the end of a literature review.

Reply: Thank you for your comment. In response, we have revised the discussion and conclusion sections to more clearly highlight the main findings, their implications for practice and policy, and the need for future research

Comment: There are also several issues in formatting and language use. For example, line 59 contains two periods, and line 64 refers unclearly to “this figure", which should be Figure 1. Abbreviations are not consistently introduced or defined. A list of abbreviations would aid readability, and those only used once or twice may not require abbreviation at all. Schemes 1 and 2 are there, but it is not clear whether these are figures or tables.

Reply: Thank you for your feedback. We have corrected the double period in line 59 and clarified all references to figures and tables All abbreviations are now defined at first use. We also standardized the labeling and referencing of schemes to tables throughout the manuscript.

Reviewer 3 Report

Comments and Suggestions for Authors

The manuscript is a narrative review examining the prevalence, risk factors and outcomes of moral injury among medical personnel and first responders in various healthcare emergency response settings in studies carried out between 2010 and 2025.

  1. Abbreviations should be explained before first use – row 40 – MI
  2. Rows 40-42 – The paragraph is unclear (in these settings....). Also one reference is not enough for such a wide range of prevalence rates.
  3. Rows 63-67. Sorry but i can’t figure out how can the authors remove 248 duplicate records (out of 227 records) and end up with 477.
  4. Rows 63-67 “three databases: PubMed, Google Scholar  and the Cochrane Central Register of Controlled Trials (CENTRAL)”. But on rows 94-98 is stated  that Four electronic databases (PubMed, CENTRAL, Google Scholar and Semantic 94 Scholar corpus) were used. Also 3 databases are mentioned in the abstract
  5. Rows 218-221 ...there is not such info in reference no 22.. Or at least, I couldn’t find any
  6. Row 210, Maguen et al. 2025 found Clinically meaningful moral injury symptoms in 4.1% of first responders and 7.3% of healthcare workers (N=1232). The study carried out by Akhtar et al., between 2020-2021 had 108 subjects facing COVID-19 Pandemic (44% had high level of moral injury). I believe that, in order to insure the quality of the research, studies that were carried out during the COVID-19 pandemic should be analyzed in a separate section and not compared per se with the other studies

Although the review generally achieved the objectives of synthesizing and consolidating knowledge about prevalence, risk factors, etc. among healthcare professionals and first responders, in terms of strategies, the review only served to identify gaps in published research, rather than to present effective solutions already demonstrated in the 41 studies.

Author Response

Thank you for reviewing our manuscript titled " Moral injury among medical personnel and first responders across different healthcare and emergency response settings" (ijerph-3658311).

Enclosed please find our response to the reviewers' comments and the revised manuscript. We hope you find our manuscript worthy of publication in MDPI.

Sincerely

Reviewer comments:

Reviewer 3#

The manuscript is a narrative review examining the prevalence, risk factors and outcomes of moral injury among medical personnel and first responders in various healthcare emergency response settings in studies carried out between 2010 and 2025.

  1. Abbreviations should be explained before first use – row 40 – MI

Reply: We have explained all Abbreviations throughot the revised manuscript.

  1. Rows 40-42 – The paragraph is unclear (in these settings....). Also one reference is not enough for such a wide range of prevalence rates.

Reply: We clarified the civilian settings, shortened the description to avoid repeating information from the results section, and added additional references to better support our claims, in lines 47-49 of the revised manuscript.

  1. Rows 63-67. Sorry but i can’t figure out how can the authors remove 248 duplicate records (out of 227 records) and end up with 477.

Reply: There was one source missing, which was now fixed in lines 69-78 of the revised manuscript.

  1. Rows 63-67 “three databasesPubMed, Google Scholar  and the Cochrane Central Register of Controlled Trials (CENTRAL)”. But on rows 94-98 is stated  that Four electronic databases (PubMed, CENTRAL, Google Scholar and Semantic 94 Scholar corpus) were used. Also 3 databases are mentioned in the abstract

Reply: As described above, one source was missing in lines 63-67. The revised manuscript corrected this.

  1. Rows 218-221 ...there is not such info in reference no 22.. Or at least, I couldn’t find any

Reply: Reference #22 was mentioned in lines 318-321 of the original manuscript. We amended the sentence to include workload and social pressure from work, as these were explicitly mentioned as predictors by Laher et al.

  1. Row 210, Maguen et al. 2025 found Clinically meaningful moral injury symptoms in 4.1% of first responders and 7.3% of healthcare workers (N=1232). The study carried out by Akhtar et al., between 2020-2021 had 108 subjects facing COVID-19 Pandemic (44% had high level of moral injury). I believe that, in order to insure the quality of the research, studies that were carried out during the COVID-19 pandemic should be analyzed in a separate section and not compared per se with the other studies

Reply: Thank you for your thoughtful suggestion regarding the analysis of studies conducted during the COVID-19 pandemic. We recognize the unique context and potential confounding factors introduced by the pandemic, which may influence the prevalence and predictors of moral injury.

While we agree that separating COVID-19-era studies could provide additional clarity, our current analysis aimed to synthesize the available literature comprehensively and maximize statistical power. Given the limited number of studies on moral injury in certain populations, further stratification by pandemic period would have substantially reduced the sample size and statistical robustness of subgroup analyses.

To address this, we have clearly indicated in the manuscript which studies were conducted during the COVID-19 pandemic in Supplementary Table 1. We also discussed the potential impact of the pandemic context on study findings in the limitations section in lines 220-240 of the revised manuscript. We then highlighted the need for future research to explore pandemic-specific effects in greater detail in the limitation section, in lines 541-548

We believe this approach maintains the integrity and interpretability of our results while transparently acknowledging the issue you raised. We appreciate your feedback and would be happy to further clarify or expand on these points in the manuscript if desired.

  1. Although the review generally achieved the objectives of synthesizing and consolidating knowledge about prevalence, risk factors, etc. among healthcare professionals and first responders, in terms of strategies, the review only served to identify gaps in published research, rather than to present effective solutions already demonstrated in the 41 studies.

Reply: Thank you for your comment. In response, we have added a dedicated paragraph to the Discussion section that explicitly addresses the current evidence for effective interventions targeting moral injury, in lines 472-484 of the revised manusciprt. This new section summarizes the limited but emerging findings from the included studies, highlights promising strategies where available, and underscores the need for further research to establish and evaluate effective interventions.

Round 2

Reviewer 2 Report

Comments and Suggestions for Authors

Thank you for resubmitting your manuscript entitled “Moral Injury among Medical Personnel and First Responders across Different Healthcare and Emergency Response Settings: A Narrative Review.” I appreciate that you have considered previous comments in this revised version. This version is considerably improved in structure and coherence, and I commend you for the progress made. Please allow me to share a few further observations that may help strengthen the manuscript.

In line 68, the term “full text removal” is unclear and should be revised or clarified. 

In line 96, the phrase “on on” appears to be a typographical or grammatical error. Similarly, the sentence structure around line 106 seems repetitive and would benefit from careful editing to improve clarity and flow.

The phrase “military relevance” in line 186 appears to be an unfinished or unclear thought. Please complete or clarify the meaning intended here.

In line 193, you refer to review articles, but whether these were included in the final analysis or excluded remains ambiguous. Please clarify their role in the selection process.

The paragraph beginning at line 222 appears to include content already presented earlier in the manuscript, suggesting redundancy. Consider condensing or removing repeated material for clarity and conciseness. Furthermore, lines 305 to 307 present observations that are difficult to interpret as concrete results. These reflections or contextual insights should be appropriately labelled as such or moved to the discussion section.

In general, I recommend expanding the subsections of Chapter 3 to present a detailed and more systematic account of the findings. The thematic or categorical structure could be better delineated to help readers follow the line of argument and identify patterns in the literature.

Finally, the text between lines 414 and 418 reads more like a conclusion than a component of the discussion section. Consider whether this content should be moved or reformulated to better distinguish between interpretive discussion and final summary remarks.

With these improvements, your manuscript will gain further clarity, precision, and impact. Again, thank you for your efforts and for carefully addressing earlier comments.

Author Response

Dear Reviewer,

Thank you very much for your thoughtful and encouraging comments regarding our revised manuscript, “Moral Injury among Medical Personnel and First Responders across Different Healthcare and Emergency Response Settings: A Narrative Review.” We greatly appreciate your recognition of the improvements made in the structure and coherence of the manuscript.

We are grateful for your continued guidance and for sharing further observations to help us strengthen our work. Your insights are invaluable, and we look forward to addressing your suggestions to enhance the quality of our submission.

Comment 1: "In line 68, the term “full text removal” is unclear and should be revised or clarified."

Response 1: The term "full text retrieval" was changed to "leaving 54 reports for further investigation" in lines 67-68 of the revised manuscript

Comment 2: "In line 96, the phrase “on on” appears to be a typographical or grammatical error. Similarly, the sentence structure around line 106 seems repetitive and would benefit from careful editing to improve clarity and flow.

Response 2: line 96 was corrected, and lines 106-108 were clarified.

Comment 3: "The phrase “military relevance” in line 186 appears to be an unfinished or unclear thought. Please complete or clarify the meaning intended here."

Response 3: This was clarified to: "Military relevance refers to any aspect of the study that pertains to military service, including populations with military backgrounds, settings within military institutions, or outcomes applicable to military personnel," in lines 186-188 of the revised manuscript.

Comment 4: "In line 193, you refer to review articles, but whether these were included in the final analysis or excluded remains ambiguous. Please clarify their role in the selection process.

Response 4: We added "included in this review" at the beginning of this sentence for clarification, in lines 199-200 of the revised manuscript.

Comment 5: "The paragraph beginning at line 222 appears to include content already presented earlier in the manuscript, suggesting redundancy. Consider condensing or removing repeated material for clarity and conciseness."

Response 5: Thank you for pointing out the redundancy in the paragraph beginning at line 222. In response to your comment, we have carefully revised lines 207–228 of the manuscript to condense the content and eliminate repetition. The revised section is now more concise and focused, improving both clarity and readability.

Comment 6: "Furthermore, lines 305 to 307 present observations that are difficult to interpret as concrete results. These reflections or contextual insights should be appropriately labelled as such or moved to the discussion section."

Response 6:  Thank you for your observation. The content from lines 305 to 307 has been relocated to lines 367–368 in the Discussion section, where it is now appropriately presented as contextual insight.

Comment 7: "In general, I recommend expanding the subsections of Chapter 3 to present a detailed and more systematic account of the findings. The thematic or categorical structure could be better delineated to help readers follow the line of argument and identify patterns in the literature."

Response 7: Thank you for the helpful feedback. We have clarified and reinforced the thematic structure of Chapter 3 through minimal but focused revisions. These include a new introductory paragraph outlining the organizational framework, clearer subsection titles, and brief transitions to enhance coherence. These updates aim to help readers more easily identify key patterns and thematic groupings in the findings while preserving the original content and structure.

Comment 8: "Finally, the text between lines 414 and 418 reads more like a conclusion than a component of the discussion section. Consider whether this content should be moved or reformulated to better distinguish between interpretive discussion and final summary remarks."

Response 8: Thank you for the comment. We revised the final lines of the Discussion to remove the summary-style conclusion and replaced them with a more interpretive closing sentence. No changes were made to the Conclusion section, as it already clearly summarizes the key findings and implications.

Reviewer 3 Report

Comments and Suggestions for Authors

all issues were addressed and corrected, although some references to row lines are incorrect (472- or 521-). final text should be revised in order to enhance clarity 

Author Response

Thank you for your feedback. We appreciate your acknowledgment that the issues were addressed. We have reviewed the final text again and made minor edits to improve clarity throughout. We also corrected the outdated line number references in the response document to ensure consistency with the revised manuscript.